# Spatiotemporal Dynamics and Influencing Factors of Vegetation Net Primary Productivity in the Yangtze River Delta Region, China

Tinghui Wang [1,†], Mengfan Gao [2,†], Qi Fu [1,3,4,*] and Jinhua Chen [1,3,4,*]

1 School of Politics and Public Administration, Soochow University, Suzhou 215123, China; 20204002013@stu.suda.edu.cn
2 School of Urban and Rural Construction, Shanxi Agricultural University, Jinzhong 030801, China; gaomf@sxau.edu.cn
3 The Institute of Regional Governance, Soochow University, Suzhou 215123, China
4 Research Institute of Metropolitan Development of China, Soochow University, Suzhou 215123, China
* Correspondence: fuqi@suda.edu.cn (Q.F.); jhchen@suda.edu.cn (J.C.)
† These authors contributed equally to this work.

**Abstract:** Vegetation Net Primary Productivity (NPP) plays a crucial role in terrestrial carbon sinks and the global carbon cycle. Investigating the spatiotemporal dynamics and influencing factors in the Yangtze River Delta (YRD) region can furnish a solid scientific foundation for green, low-carbon, and sustainable development in China, as well as a reference for other rapidly urbanizing regions. This study focuses on the YRD region as an illustration and utilizes the Carnegie–Ames–Stanford Approach (CASA model) to quantify NPP in this region from 2000 to 2018. Investigation into the spatiotemporal dynamics and influencing factors was conducted using Theil–Sen median trend analysis and scenario analysis. The results indicate that the NPP in the YRD region from 2000 to 2018 exhibited pronounced spatial differentiation characteristics, typically exhibiting a spatial distribution pattern of being high in the south and low in the north, high in the west and low in the east. Additionally, the expansion of built-up areas and the reduction in cultivated land have the potential to reduce NPP in the YRD region. Moreover, the influence of land-use and land-cover change (LULCC) is anticipated to be relatively limited compared to that of climate change. Furthermore, changes in precipitation were found to be positively correlated with changes in NPP, with the effect being relatively more pronounced. The correlation between temperature and NPP demonstrated spatial differentiation, with a mainly positive correlation in the central and southern parts of the YRD and a mainly negative correlation in the northern part. Changes in solar radiation had a negative correlation with changes in NPP. Based on these results, it is recommended that local governments strictly enforce urban development boundaries and manage the disorderly expansion of built-up areas, enhance the regional irrigation infrastructure, and address air pollution, so as to ensure the necessary conditions for the growth of vegetation, reduce greenhouse gas emissions, and control regional temperature rises. This study can provide stronger evidence for revealing the influencing mechanisms of NPP through the control of impact conditions and the exclusion of confounding factors via scenario analysis. The policy implications can offer insights into NPP enhancement and environmental management for the YRD and other rapidly urbanizing regions.

**Keywords:** NPP; spatiotemporal dynamics; influencing factors; scenario analysis; Yangtze River Delta region





## 1. Introduction

Since the Industrial Revolution, the rapid development of the global economy and the significant utilization of diverse energy sources have led to the emission of substantial amounts of greenhouse gases, such as $CO_2$, $CH_4$, and $N_2O$. Notably, the rise in $CO_2$ levels

has contributed to a gradual acceleration in the global average temperature over the last 200 years [1]. Terrestrial ecosystems have a robust capacity for carbon sequestration and can sequester atmospheric $CO_2$ as organic compounds through photosynthesis in vegetation, consequently playing a pivotal role in mitigating anthropogenic carbon emissions and contributing to the global carbon cycle [2,3]. Vegetation Net Primary Productivity (NPP) is defined as the net quantity of photosynthetic organic matter taken in per unit area of vegetation over a specific period, showcasing the vegetation's capability to sequester and convert carbon, and serving as a fundamental element of terrestrial carbon sinks and the global carbon cycle [4–6]. Numerous scholars posit that assessing NPP not only characterizes the physiological and ecological condition of vegetation but also mirrors alterations in its carbon sequestration capacity through spatial and temporal transformations [7,8]. Simultaneously, the analysis of NPP response to diverse factors is essential, as it can offer scientific backing for augmenting carbon sinks and advancing the carbon cycle.

The assessment of NPP at various scales globally primarily involves site measurements and model simulation methods. Site measurements allow for the quantification of carbon sequestration in vegetation and soils through the collection of field samples. For instance, in a study conducted by Fang et al. [9], over 350 researchers systematically assessed carbon stocks and their spatial distribution in China's terrestrial ecosystems, examining over 17,000 sample plots and compiling 200,000 data points. While the outcomes of these assessments are reasonably precise, the substantial investment in human and material resources also constrains the extensive application of large-scale field measurements [10]. The recent advancements in remote sensing monitoring technology have enhanced the accessibility of parameters like precipitation, temperature, and light energy utilization ratio. Notably, these methods encompass the Global Production Efficiency Model [11], the Eddy Covariance-LUE Model [12], and the Carnegie–Ames–Stanford Approach (CASA model) [13]. The seamless integration of the CASA model with RS and GIS, coupled with the ease of parameter acquisition, establishes it as a prevalent model known for its high measurement accuracy [6]. For instance, certain researchers conducted studies on NPP assessment in the Tibetan Plateau [14] and central China [15]. Furthermore, non-parametric tests like the Theil–Sen median trend analysis and the Mann–Kendall test, which analyze spatial trends in time series data, are extensively employed due to their robust statistical foundations for evaluating significance levels [16,17].

The spatiotemporal dynamics of NPP are primarily shaped by natural and anthropogenic factors [18]. Among these, climate change, and land-use and land-cover change (LULCC) are regarded as pivotal influencing factors. For instance, research studies have shown that precipitation, temperature, and solar radiation can impact vegetation growth [19–21]. Comprehending the individual and combined effects of climate change and LULCC on terrestrial ecosystem productivity is crucial for sustainable ecosystem management globally and regionally [22,23]. Nevertheless, due to the interactive nature of climate change and LULCC and the intricate and interconnected effects on vegetation [24,25], investigating the separate impacts of LULCC and climate change on NPP presents a significant challenge [26]. As most past studies relying on correlation analyses find it challenging to separate the impacts of other variables, some scholars have turned to scenario analysis to differentiate the influences of LULCC and climate change on NPP, offering a viable approach for such investigations. For instance, Fu et al. developed three scenarios to assess how LULCC and climate change affected water yield, soil conservation, crop production, and sand stabilization in the Altai region over each decade from 1990 to 2010 [27].

The Yangtze River Delta (YRD) region boasts ample rainfall and optimal temperature conditions, positioning it as a high-intensity carbon sink functional area in China [28]. Additionally, being affected by the subtropical monsoon, the region shows heightened sensitivity to climate variations [29]. Moreover, the area serves as a paradigm for the swift urbanization and industrialization occurring in China, with LULCC exerting a substantial influence on NPP. Thus, investigating the YRD region can offer a significant reference for

unveiling the NPP change patterns and the underlying influence mechanism. Furthermore, with China's commitment to achieving carbon neutrality and peak carbon dioxide emissions, coupled with the implementation of integrated eco-friendly development in the YRD region, investigating the spatiotemporal dynamics and influencing factors in the YRD region can establish a scientific foundation for green, low-carbon, and sustainable development in China, and serve as a blueprint for other fast-urbanizing areas.

Therefore, this investigation will use the YRD region as a case study and employ the CASA model to assess NPP in this area from 2000 to 2018. The spatiotemporal dynamics and influencing factors will be examined using Theil–Sen Median trend analysis and scenario analysis. The main research objectives of this paper encompass three key areas: (1) exploring the spatiotemporal dynamics of NPP in the YRD region from 2000 to 2018; (2) unveiling the impact of LULCC and climatic factors on NPP in the YRD region; (3) proposing specific recommendations to enhance NPP levels in the YRD region.

This study analyses the impact of LULCC and climatic factors on NPP by the control of impact conditions and the exclusion of confounding factors through scenario analysis, so as to provide stronger evidence for revealing the influencing mechanism of NPP. The practical value is to provide the spatiotemporal dynamics of NPP in the YRD region and the policy implications for enhancing the level of NPP, which in turn serves as a reference for environmental management in the YRD region and other rapidly urbanizing regions.

## 2. Materials and Methods

### 2.1. Study Area

The Yangtze River Delta region is situated on the eastern coastline of China, encompassing the lower course of the Yangtze River, characterized by an alluvial plain predating its discharge into the sea (see Figure 1a). Encompassing Anhui Province, Jiangsu Province, Zhejiang Province, and Shanghai Municipality, the region comprises 41 cities and covers an area of 358,000 square kilometers, representing approximately 3.7% of China's total land area. The area experiences plentiful rainfall and features sophisticated water infrastructure, with an average annual precipitation ranging between 704 and 2000 mm, and average annual temperatures varying from 12.2 to 18.9 °C. The climatic conditions in the YRD region are primarily characterized by a subtropical monsoonal climate, with the vegetation predominantly consisting of subtropical broad-leaved evergreen forests.

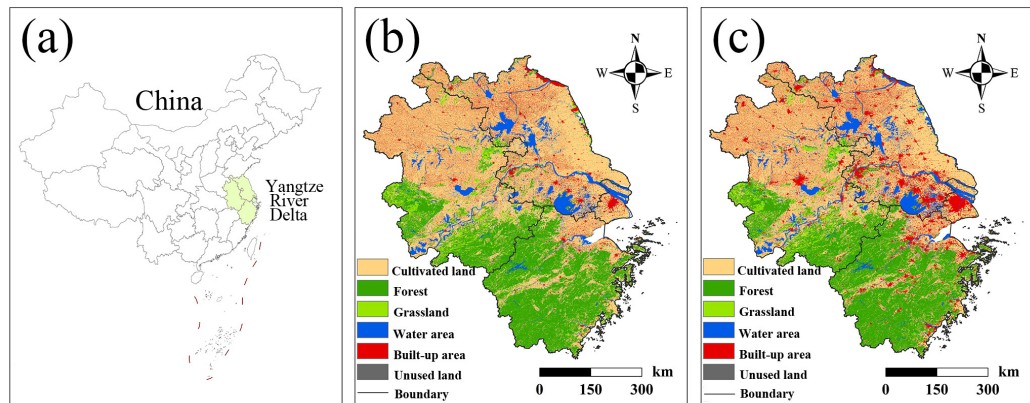

**Figure 1.** Map of regional location and land-use types in the YRD region: (**a**) regional location, (**b**) land-use types in 2000, (**c**) land-use types in 2018.

The YRD region exemplifies a typical area characterized by rapid urbanization and industrialization in China, boasting an average urbanization rate of 75.01%, surpassing the national average by 11.12 percentage points. By 2022, the resident population in the region has surged to 237 million, representing approximately 16.8% of the total national population, demonstrating a strong ability to attract population. Concurrently, the YRD region stands as one of the most dynamic areas in China in terms of economic advancement.

In 2022, the region's GDP surged to 29.03 trillion yuan, marking a 1.8-fold increase from 2015 and representing almost a quarter of the nation's GDP. Furthermore, the secondary sector's portion of the GDP surpassed 40% in all three provinces.

In conclusion, the YRD region offers favorable natural conditions conducive to vegetation growth, serving as a foundation to elucidate the impact of climate change on NPP. Simultaneously, anthropogenic activities like urban expansion, population agglomeration, and industrialization can contribute to regional changes in LULCC, consequently disrupting NPP. Hence, selecting the YRD region as the study area facilitates the examination of the effects of LULCC and climate change on NPP, offering valuable insights for other swiftly urbanizing areas.

### 2.2. Data Sources

The data essential for NPP measurement and analysis of influencing factors are sourced from the Normalized Difference Vegetation Index (NDVI) data, meteorological data, and LULCC data (see Table 1). Specifically, the NDVI data primarily serves for NPP measurement, while LULCC data and meteorological factors, including total monthly precipitation, average monthly temperature, and total monthly sunshine hours, are predominantly utilized for both NPP measurement and analysis of influencing factors. The dataset spans five key years: 2000, 2005, 2010, 2015, and 2018.

**Table 1.** Research data and sources.

| Data Name | Data Type | Data Source |
| --- | --- | --- |
| NDVI | 1 km × 1 km raster data | Resource and Environment Science and Data Center (http://www.resdc.cn/Default.aspx, accessed on 1 March 2023) |
| LULCC | 30 m × 30 m raster data | Resource and Environment Science and Data Center (http://www.resdc.cn/Default.aspx, accessed on 2 March 2023) |
| Total monthly precipitation | Site data | China Meteorological Data Network (http://data.cma.cn/, accessed on 7 March 2023) |
| Average monthly temperature | Site data | China Meteorological Data Network (http://data.cma.cn/, accessed on 9 March 2023) |
| Total monthly sunshine hours | Site data | China Meteorological Data Network (http://data.cma.cn/, accessed on 10 March 2023) |

The NDVI data and LULCC data were processed in ArcGIS 10.2, resampled into 100 m × 100 m raster data. Monthly sunshine hours data were calculated to derive the total monthly solar radiation data for each station in the YRD region using the "calculate total monthly solar radiation at station" tool in the Zhu Wenquan extension module within ENVI. Total monthly precipitation, average monthly temperature, and monthly solar radiation site data were derived as 100 m × 100 m raster data using kriging interpolation in ArcGIS. Both vector and raster data were transformed to the identical projected coordinate system (Krasovsky_1940_Albers). Subsequently, the precipitation, temperature, solar radiation, NDVI, and LULCC datasets were cropped into layers with consistent rows and columns for each month spanning the years 2000, 2005, 2010, 2015, and 2018.

### 2.3. Methods

The analytical workflow of this study comprised three main steps (Figure 2). Initially, NPP was quantified using the CASA model, with results validated against related studies and MOD17A3 data. Subsequently, the spatial distribution and dynamic trends of NPP were investigated through trend analysis. Secondly, the spatiotemporal variations in LULCC and climate factors, including precipitation, temperature, and solar radiation, were analyzed. Concurrently, four simulated scenarios were established to investigate the effects of various

factors on NPP using trend analysis. Thirdly, the NPP changes in the simulated scenarios were compared to the realistic scenario. By combining the spatiotemporal variations of LULCC and climate factors, the impacts of LULCC and climate change on NPP were discussed. Based on this analysis, policy implications for enhancing NPP levels in the YRD region were suggested.

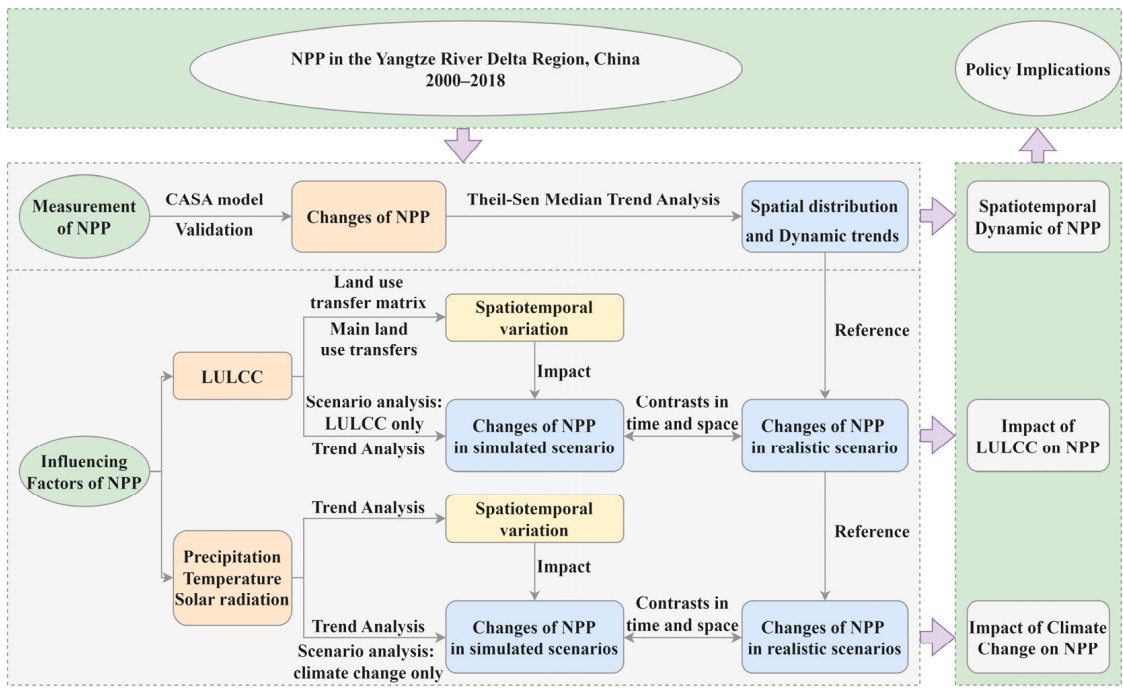

**Figure 2.** The flow chart of this study.

### 2.3.1. CASA Model

This study builds upon Zhu Wenquan's enhanced CASA model [30–33], utilizing the corresponding module in ENVI for calculating the NPP of the YRD region spanning the period from 2000 to 2018. The methodological framework of the CASA model is outlined as follows:

Measurement of NPP

$$NPP(x) = \sum_{t=1}^{12} NPP(x, t) \times S_p \tag{1}$$

In this formula, $NPP(x)$ represents the full year NPP of image element x (g, in C); $NPP(x, t)$ represents the unit area NPP in image element x for month t (g·m$^{-2}$·mon$^{-1}$); and $S_p$ represents the area of the image element (m$^2$).

The NPP measured in the model can be represented by two key factors: the Absorbed Photosynthetic Active Radiation (APAR) and the actual light energy utilization ratio ($\varepsilon$) of the plant.

$$NPP(x, t) = APAR(x, t) \times \varepsilon(x, t) \tag{2}$$

where $APAR(x, t)$ denotes the photosynthetically active radiation absorbed by image x in month t (MJ·m$^{-2}$) and $\varepsilon(x, t)$ denotes the actual light energy utilization ratio of image x in month t (MJ·m$^{-2}$).

Calculation of APAR

Photosynthetically active radiation (PAR, 0.4–0.7 µm) acts as the primary driving force in plant photosynthesis. Utilizing the reflectance characteristics of vegetation in the infrared and near-infrared bands enables remote sensing data to accurately estimate the fraction

of PAR absorbed by plant foliage (APAR). The APAR absorbed by a plant is dependent on both the total solar radiation and the plant's inherent characteristics. The formula is as follows:

$$APAR(x, t) = SOL(x, t) \times FPAR(x, t) \times 0.5 \tag{3}$$

In this formula, $SOL(x, t)$ denotes the total solar radiation ($MJ \cdot m^{-2}$) for month t at image element x; $FAPAR(x, t)$ indicates the proportion of the incident photosynthetically active radiation absorbed by the vegetation layer at image element x for month t; and the constant "0.5" signifies the proportion of total solar radiation that is utilized by vegetation as effective solar radiation (wavelengths of 0.38–0.71 μm).

Calculation of FPAR

NDVI can reflect the ground surface vegetation cover and therefore can be used to calculate the fraction of absorbed photosynthetically active radiation (FPAR). The formula is given below:

$$FPAR(x, t)_{NDVI} = \frac{NDVI(x, t) - NDVI_{i,min}}{NDVI_{i,max} - NDVI_{i,min}} \times (FPAR_{max} - FPAR_{min}) + FPAR_{min} \tag{4}$$

$$FPAR(x, t)_{SR} = \frac{SR(x, t) - SR_{i,min}}{SR_{i,max} - SR_{i,min}} \times (FPAR_{max} - FPAR_{min}) + FPAR_{min} \tag{5}$$

In this formula, $FPAR_{max}$ and $FPAR_{min}$ are 0.95 and 0.001; $SR_{i,max}$ and $SR_{i,min}$ denote the maximum and minimum values of the simple ratio index for vegetation type i, which correspond to the 95% and 5% lower percentile of the NDVI for vegetation type i, respectively. $SR(x, t)$ can be calculated by the following formula:

$$SR(x, t) = \left[ \frac{1 + NDVI(x, t)}{1 - NDVI(x, t)} \right] \tag{6}$$

The more common way to reduce the error is to calculate the mean of the two. The formula is as follows:

$$FPAR(x, t) = \frac{[FPAR(x, t)_{NDVI} + FPAR(x, t)_{SR}]}{2} \tag{7}$$

In the formula, $FPAR(x, t)_{NDVI}$ and $FPAR(x, t)_{SR}$ denote the results estimated from NDVI and SR, respectively.

Calculation of the actual light energy utilization ratio (ε)

The light energy utilization ratio is defined as the ratio of the chemical potential embedded in the dry matter produced per unit area over a given period to the photosynthetically active radiant energy projected onto the same area. Environmental factors, including temperature, soil moisture, and atmospheric pressure variations, regulate NPP by influencing the photosynthetic capacity of plants. This regulatory effect on NPP by such factors is achieved through revising maximum light energy utilization within the remote sensing model. In an ideal scenario, vegetation achieves maximum light energy utilization; however, under realistic conditions, this is primarily influenced by temperature and moisture. The formula is as follows:

$$\varepsilon(x, t) = T_{\varepsilon 1}(x, t) \times T_{\varepsilon 2}(x, t) \times W_{\varepsilon}(x, t) \times \varepsilon_{max} \tag{8}$$

In this formula, $T_{\varepsilon 1}(x, t)$ and $T_{\varepsilon 2}(x, t)$ represent the stress effects of low and high temperatures on light energy utilization, respectively (unitless); $W_{\varepsilon}(x, t)$ denotes the coefficient of water stress (unitless), indicating the impact of water conditions; $\varepsilon_{max}$ signifies the maximum light energy utilization under ideal conditions ($g(C) \cdot MJ^{-1}$). For the maximum light energy utilization, static parameters are referenced from Zhang et al. [26] for calculating the

research results related to the NPP of the Yangtze River Basin. Additionally, the maximum light energy utilization in areas without vegetation is assigned a value of 0 (Table 2).

**Table 2.** Maximum light energy utilization for each land type in the YRD region ($\varepsilon_{max}$).

| Code | Land Type | $\varepsilon_{max}$ |
|:---:|:---:|:---:|
| 1 | Cultivated land | 0.95 |
| 2 | Forest | 1.105 |
| 3 | Grassland | 0.788 |
| 4 | Water area | 0 |
| 5 | Built-up area | 0 |
| 6 | Unused land | 0 |

2.3.2. Theil–Sen Median Trend Analysis

Utilizing Theil–Sen median trend analysis and the Mann–Kendall test, this study reveals the spatial and temporal changes and influencing factors of NPP in the YRD region. Theil–Sen median trend analysis is based on calculating the median of the slopes of $n(n-1)/2$ combinations of data and numerous studies have established this method as reliable for temporal estimates [34–36]. The formula is as follows:

$$\beta = \text{median}\left[\frac{\Delta y}{\Delta t}\right] \tag{9}$$

In this formula, the Slope of Sen ($\beta$) represents the median of all slopes calculated between every consecutive data point in the NPP time series (x); $\Delta y$ represents the change in image element NPP due to time variation; and $\Delta t$ represents the change in image element time. When the Slope of Sen's result is greater than 0, it indicates an increasing trend in NPP; conversely, a negative value indicates a decreasing trend in NPP. The Mann–Kendall mutation test stands as a non-parametric, longitudinal time series analysis method, attributed to H.B. Mann [37] and M.G. Kendall [38]. Unlike parametric trend tests, this method does not require the data to follow a normal distribution. Instead, it requires the data to be independent and resilient to outliers in time series analysis. The formula is as follows:

$$S = \sum_{i=1}^{n} \sum_{j=i+1}^{n} \text{sign}(x_i - x_k) \tag{10}$$

$$\text{sgn}(x_i - x_k) = \begin{cases} +1 & (x_i - x_k) > 0 \\ 0 & (x_i - x_k) = 0 \\ -1 & (x_i - x_k) < 0 \end{cases} \tag{11}$$

$$\text{Var}(S) = \frac{s(n-1)(2n+5)}{18} \tag{12}$$

$$Z = \begin{cases} \frac{S-1}{\sqrt{\text{Var}(S)}} & (S > 0) \\ 0 & (S = 0) \\ \frac{S+1}{\sqrt{\text{Var}(S)}} & (S < 0) \end{cases} \tag{13}$$

In this formula, S represents the value of the statistical variable, n represents the sample size, and $x_i$ and $x_k$ represent the time series data; sgn represents the sign function and Z represents the trend direction. When the Z value is negative, it means that the NPP of the image element shows a decreasing trend during the study period and, vice versa, an increasing trend. Moreover, if the absolute value of Z exceeds 1.96 within the 95% confidence interval, the original hypothesis of no trend over the study period will be rejected for that image element's NPP. We performed calculations based on the R 4.0.3 software platform and obtained the trend changes in each layer through the raster calculator in ArcGIS.

Referring to the research ideas presented in prior studies [39], we characterized the outcomes derived from Theil–Sen median trend analysis into three distinct categories: values exceeding 0.0005 were interpreted as indicating an increase, those falling within the range of −0.0005 to 0.0005 were considered as reflecting stability, and any values below −0.0005 were deemed to signify a decrease. The Mann–Kendall test serves as a means of validating the results obtained from Theil–Sen median trend analysis by assessing the significance associated with each raster in the analysis. Subsequently, in cases where the absolute Z-score is less than −1.96 or greater than 1.96 at a significance level of $p < 0.05$, it signifies a 95% confidence level for statistical significance. Finally, we employed the reclassification tool and raster calculator within ArcGIS to categorize the patterns of change into distinct classes, including significant increase, increase, stability, decrease, or significant decrease.

2.3.3. Scenario Analysis

Scenario analysis allows for the focused examination of particular influencing factors while mitigating the impact of extraneous variables on experimental outcomes. In this study, we draw upon insights from previous research [26,27] to investigate the impact of various scenarios on the influencing factors affecting NPP levels. The scenarios are examined from 2000 to 2018, with a spatial resolution of 100 m × 100 m serving as the foundational unit for analysis. Four distinct scenarios (refer to Table 3) were crafted to account for LULCC and the impact of three climate factors. These influencing factors were sequentially altered for the years 2000, 2005, 2010, 2015, and 2018, followed by a comparison with NPP levels under realistic scenarios for each year.

**Table 3.** Scenario design for influencing factors.

| Group | Scenario | LULCC | NDVI | Precipitation | Temperature | Solar Radiation |
|---|---|---|---|---|---|---|
| LULCC | L1 | ▲ | △ | △ | △ | △ |
| | L2 | △ | △ | ▲ | △ | △ |
| Climate change | L3 | △ | △ | △ | ▲ | △ |
| | L4 | △ | △ | △ | △ | ▲ |

Note: Black triangles (▲) represent factors that changed, white triangles (△) represent factors that remained constant.

## 3. Results

### 3.1. Validation of the CASA Model

Through two distinct approaches, we validated the measurement results of the CASA model. Firstly, we compared the measurement results of the CASA model presented in this study with those from related studies. Upon comparing similar temporal and spatial scales, we found that the NPP results measured in this study and those obtained in related studies exhibited substantial similarity (Table 4). Secondly, the MOD17A3 data were processed to calculate the annual mean NPP value for the YRD region. Upon comparing the CASA model's results with these, we observed that both methods produced highly similar NPP levels (Table 5). Consequently, the reliability of the NPP measurements for the YRD region obtained using the CASA model in this study is validated, permitting further analysis.

**Table 4.** Validation compared to findings of relevant studies.

| Validation Reference | Measured Area | Time (Year) | NPP (gC/(m²·a)) | NPP of This Study (gC/(m²·a)) |
|---|---|---|---|---|
| CASA model [40] | Core city clusters in the YRD | 2000–2016 | 419.33 | 510.10 |
| Based on MOD17A3 [41] | YRD | 2000–2019 | 550.17 | 538.69 |
| CASA model [42] | Zhejiang Province | 2006 | 625.68 | 583.69 |
| Based on MOD17A3 [43] | Jiangsu Province | 2000–2006 | 506.6 | 525.58 |

**Table 5.** Validation compared to the MOD17A3 data (gC/(m$^2$·a)).

| Time (Year) | 2000 | 2005 | 2010 | 2015 | 2018 |
|---|---|---|---|---|---|
| NPP of MOD17A3 | 533.24 | 510.37 | 532.24 | 582.39 | 587.51 |
| NPP of this study | 638.54 | 583.69 | 591.29 | 669.18 | 636.39 |

*3.2. Changes in NPP*

3.2.1. Spatial Distribution of NPP

According to the measurement results of the CASA model, the variation in NPP in the YRD region from 2000 to 2018 ranged from 0 to 1776.00 gC/(m$^2$·a) at an image element scale of 100 m × 100 m. Spatially, NPP within the YRD region exhibits distinct spatial differentiation characteristics, typically revealing a spatial distribution pattern characterized by higher values in the south and lower in the north, and higher in the west and lower in the east (Figure 3). Higher NPP values (>986 gC/(m$^2$·a)) are observed in the southwestern part of Anhui Province and the majority of Zhejiang Province. Lower NPP values (<286 gC/(m$^2$·a)) are predominantly located in southern Jiangsu and Shanghai. In the northern Anhui and Jiangsu Provinces, NPP generally falls within the lower to middle range. Temporally, areas exhibiting lower NPP values have been expanding annually, predominantly dispersing and extending outward; in contrast, zones with higher values aggregate in extensive areas, demonstrating a consistent upward trend. Among these, zones with lower NPP values in the Eastern YRD experienced rapid expansion between 2000 and 2010, followed by relative stability from 2010 to 2018. Zones with lower NPP values in the central and northern YRD remained generally stable until 2010, after which they commenced a rapid expansion. In the southern and western YRD, zones with higher NPP values saw a marked increase from 2005 to 2018.

3.2.2. Dynamic Trends in NPP

The average NPP value in the YRD region exhibited a fluctuating trend between 2000 and 2018. The average NPP values for 2000 (638.54 gC/(m$^2$·a)) and 2018 (636.39 gC/(m$^2$·a)) were closely comparable (Figure 4). Average NPP values were highest in Zhejiang Province (820.53 gC/(m$^2$·a)), followed by Anhui Province (624.15 gC/(m$^2$·a)), then Jiangsu Province (458.32 gC/(m$^2$·a)), and lowest in Shanghai City (287.68 gC/(m$^2$·a)). Among these, Zhejiang Province's average NPP value consistently surpassed the YRD region's overall average (623.82 gC/(m$^2$·a)). The average NPP values are more comparable between Anhui Province and the YRD region, whereas those for Jiangsu Province and Shanghai Municipality fall below the YRD region's average. Between 2000 and 2018, average NPP values in Jiangsu Province and Shanghai City exhibited a fluctuating downward trend, with the most pronounced decrease occurring in Shanghai City (172.20 gC/(m$^2$·a)). For Anhui and Zhejiang provinces, average NPP values demonstrated a fluctuating upward trend, with significant increases notably evident between 2010 and 2015.

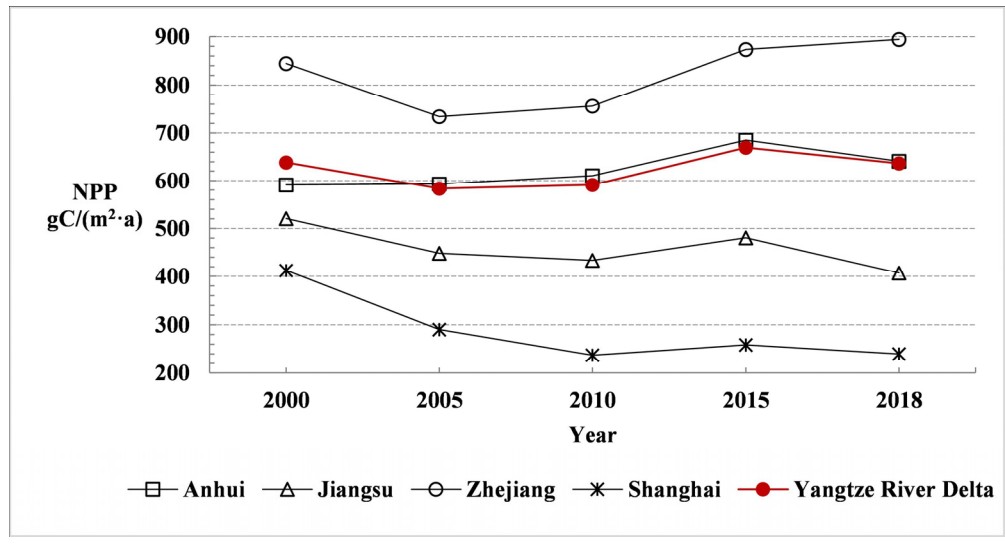

**Figure 3.** Spatial distribution of NPP in the YRD region from 2000 to 2018: (**a**) 2000, (**b**) 2005, (**c**) 2010, (**d**) 2015, (**e**) 2018.

**Figure 4.** Changes in the average NPP value of the YRD region between 2000 and 2018.

The results of the Theil–Sen median trend analysis afford further insights into the trend in NPP in the YRD region between 2000 and 2018 (Figure 5). In the YRD region, 5.98% showed a significant increase in NPP, 40.61% experienced an increase, 7.26% underwent

a significant decrease, 45.77% saw a decrease, and only 0.38% remained stable. From a spatial perspective, areas experiencing a significant increase in NPP were predominantly located in northern Anhui Province, while those with an increase were mainly found in northern Anhui, northern Jiangsu, and western Zhejiang Provinces. Furthermore, areas exhibiting both significant decreases and general reductions in NPP were primarily situated in Shanghai, central and southern Jiangsu, southern Anhui, and eastern Zhejiang Provinces.

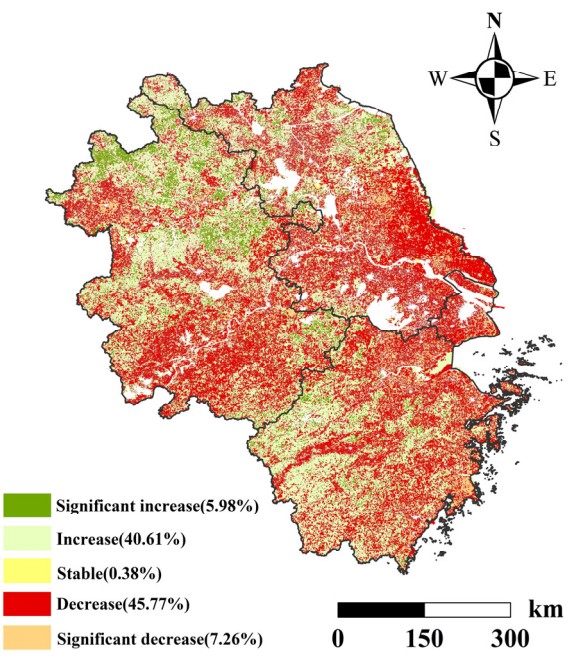

**Figure 5.** Trends in NPP of the YRD region between 2000 and 2018.

*3.3. LULCC and Scenario Analysis*

3.3.1. LULCC in the YRD Region

In terms of LULCC, between 2000 and 2018, cultivated lands, forests, and grasslands in the YRD region saw decreases. Among these changes, cultivated lands were primarily transformed into built-up areas, forests into cultivated lands, and grasslands into forests. There was a notable expansion of built-up areas, accompanied by the conversion of 16,928 km$^2$ of built-up lands back into cultivated lands (Table 6). Simultaneously, water bodies and areas of unused land experienced growth, with the former predominantly transforming into cultivated land and the latter into forests. Overall, LULCC in the YRD region from 2000 to 2018 was primarily characterized by the expansion of built-up areas and the contraction of cultivated lands.

**Table 6.** Land-use transfer matrix in YRD region, 2000–2018 (km$^2$).

| 2000–2018 | Cultivated Land | Forest | Grassland | Water Area | Built-Up Area | Unused Land | Original Area |
|---|---|---|---|---|---|---|---|
| Cultivated land | 129,535 | 13,622 | 1860 | 6312 | 31,307 | 75 | 182,711 |
| Forest | 13,544 | 79,769 | 3895 | 1124 | 2369 | 56 | 100,757 |
| Grassland | 1894 | 4001 | 4911 | 703 | 389 | 8 | 11,906 |
| Water area | 5209 | 931 | 565 | 16,233 | 1852 | 188 | 24,978 |
| Built-up area | 16,928 | 861 | 185 | 1697 | 11,718 | 7 | 31,396 |
| Unused land | 14 | 39 | 1 | 3 | 6 | 5 | 68 |
| Area after transformation | 167,124 | 99,223 | 11,417 | 26,072 | 47,641 | 339 | - |

Specifically, the most extensive conversion of cultivated land to built-up areas occurred in the 2000–2005 period, followed by transitions of cultivated land into water bodies and forests. During this period, all other land categories contributed more extensively to the expansion of built-up areas, whereas smaller proportions of built-up areas were reverted back to other land types. The 2005–2010 phase saw the greatest scale of reciprocal conversion between cultivated and built-up areas across the entire timeframe, marked by the most rapid expansion of built-up territories. Interconversion between cultivated and built-up areas decreased during 2010–2015 but then enlarged again from 2015 to 2018. However, the 2015–2018 period experienced the lowest net conversion from cultivated to built-up areas (Tables S1–S4 in the Supplementary Materials).

The primary types of land-use transformation within the YRD region, encompassing three provinces and one municipality from 2000 to 2018, are visually detailed in Figure 6. Overall, the dominant transition observed is from cultivated land to built-up areas. In contrast, transformations from built-up areas to cultivated land are less prevalent and mainly situated outside the urban core. Jiangsu Province stands out with the largest net area of cultivated land converted, amounting to 5080.5 km$^2$, predominantly clustered in the southern region, while the central and northern areas exhibit a scattered distribution. A noteworthy conversion from water areas to built-up spaces can be observed along the northeast coast of Jiangsu Province (Figure 6b). Moving to Zhejiang Province, a net area of 4582.46 km$^2$ of cultivated land has been successfully transformed, with the shift towards built-up areas concentrated in the northeast, central, and southeast coastal regions. Concurrently, the conversion of cultivated land to forests sporadically occurs in regions beyond northeastern Zhejiang Province. Furthermore, Zhejiang Province showcases a relatively significant amount of forest land conversions, primarily transitioning to cultivated land and built-up areas (Figure 6c). In Anhui Province, the net area of cultivated land conversion amounts to 3114.51 km$^2$, with the transition towards built-up areas mainly concentrated in urban zones. Noteworthy transformations include cases where cultivated land has been converted to forests in the mountainous western and southern regions of Anhui Province (Figure 6a). Contrasting the provincial analyses, Shanghai demonstrates a striking ratio, where cultivated land conversions into built-up areas surpass conversions from built-up areas to cultivated land by approximately 11.57 times. Additionally, cultivated land transitioning to built-up areas appears more frequently and clustered around developed regions, while conversions from built-up areas to cultivated land are sporadically distributed along the northern and southern urban peripheries (Figure 6d).

3.3.2. Scenario Analysis: LULCC Only

In the L1 scenario, the climate factor was maintained at the 2000 level, solely focusing on LULCC, thereby influencing NPP through LULCC effects. The NPP trend in the YRD region from 2000 to 2018 displays fluctuating patterns, starting with an increase, followed by a decrease, then rising again, showcasing a deviation from observed trends in realistic scenarios (Figure 7a). When considering only LULCC effects, the average NPP in 2005 rose from 2000, peaking as the highest value over five years at 660.63 gC/(m$^2$·a), surpassing the average NPP in the realistic scenario (583.69 gC/(m$^2$·a). Subsequently, there was a decline in 2010 and 2015. However, the average NPP in 2010 (645.36 gC/(m$^2$·a)) exceeded that of the realistic scenario (591.29 gC/(m$^2$·a)), whereas the average NPP in 2015 (636.04 gC/(m$^2$·a)) fell below the realistic scenario (669.18 gC/(m$^2$·a). By 2018, the average NPP documented a slight recovery at 634.67 gC/(m$^2$·a), marginally below the level of the realistic scenario (636.39 gC/(m$^2$·a).

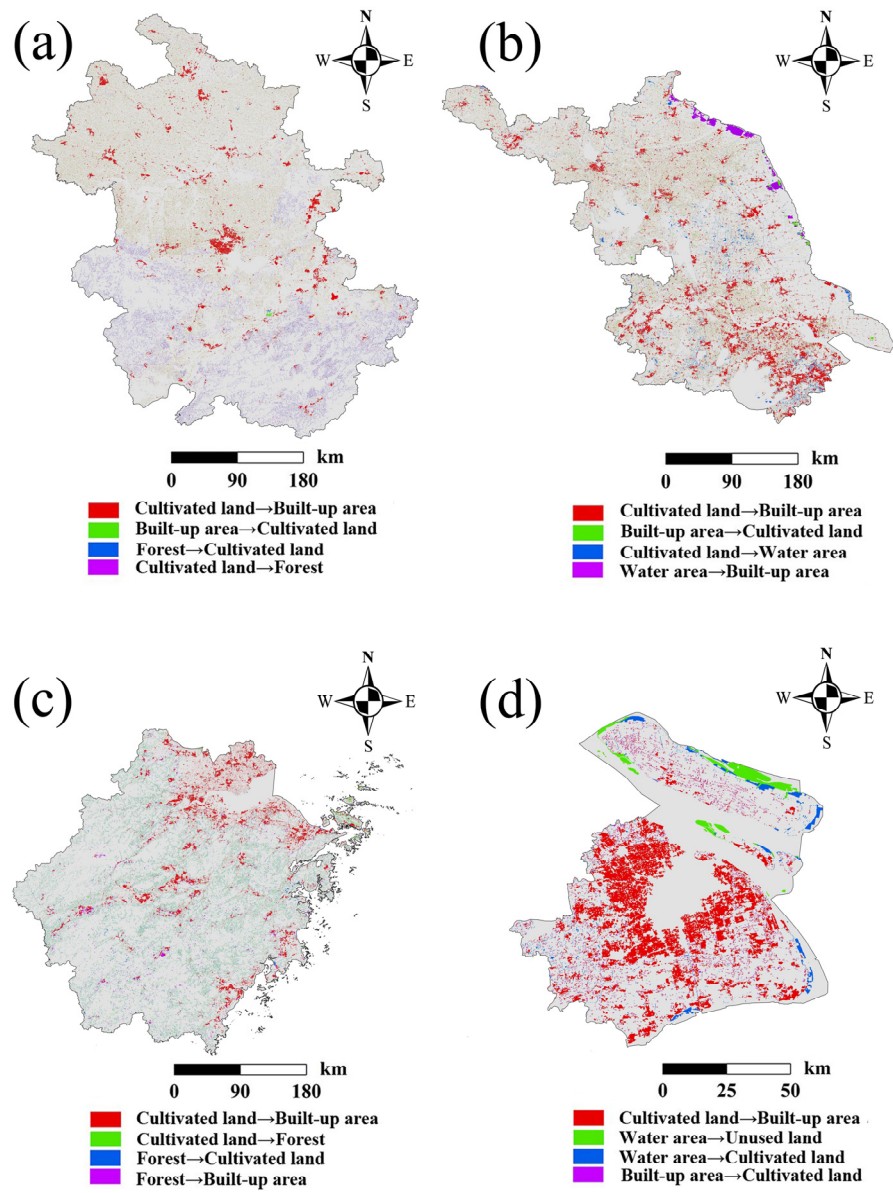

**Figure 6.** Main land-use transfers in the YRD region, 2000–2018: (**a**) Anhui Province, (**b**) Jiangsu Province, (**c**) Zhejiang Province, (**d**) Shanghai.

Analyzing NPP trends within the L1 scenario at the image element scale highlights that, under the sole influence of LULCC, NPP demonstrates an ascending trajectory in a majority of regions between 2000 and 2018, with some areas maintaining stability or experiencing a decline. It is noteworthy that there are no instances of significant NPP increase or decrease discernible in the patterns (Figure 7b). Furthermore, regions displaying an increasing NPP trend constitute 88.53% of the total, showcasing a broad distribution throughout the YRD region. Areas witnessing a decline in NPP patterns make up 9.41%, primarily characterized by clustered and sporadic distributions. In contrast, regions maintaining stability represent a mere 2.06% of the total. These trends align closely with significant land-use transitions observed in Figure 6, where regions undergoing conversion from cultivated land to built-up areas are associated with declining NPP trends. Conversely, regions experiencing growth trends exhibit spatial distributions akin to other areas, encompassing diverse land-use transitions and regions where alterations are less pronounced.

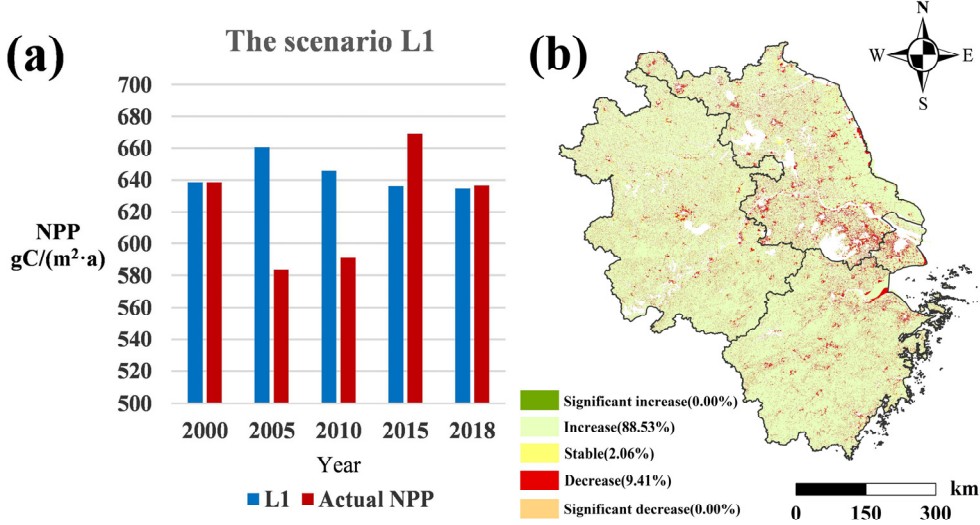

**Figure 7.** Changes in NPP in scenario L1: (**a**) comparison of L1 with the realistic scenario, (**b**) trends in NPP in L1.

*3.4. Climate Change and Scenario Analysis*

3.4.1. Climate Change in the YRD Region

Trend analyses conducted on the fluctuations of average annual precipitation within the YRD region from 2000 to 2018 reveal distinctive patterns. Regions experiencing a decline or notable decrease in average annual precipitation are concentrated in areas like northern Anhui Province and northern Jiangsu Province, accounting for 17.41% and 0.78% of the total, respectively. It is noteworthy that Zhejiang Province, Shanghai, and most other regions exhibit a rising trend in average annual precipitation, encompassing 66.80% of the total, with significant increments particularly prominent in central Anhui Province and southwestern Jiangsu Province, displaying a substantial increase of 15.01%. Temporally, the average precipitation values from 2000 to 2018 in the YRD region follow a pattern of initial increase followed by subsequent decrease. Specifically, the average annual precipitation levels were relatively low in 2000 (1231.41 mm) and 2005 (1262.60 mm), rising to a peak in 2015 (1539.73 mm), and then declining to 1339.89 mm by 2018 (Figure 8a).

Between 2000 and 2018, the YRD region demonstrates a predominant proportion of regions with a decrease or significant decrease in average annual temperature, accounting for 53.85% and 6.45%, respectively. These areas are predominantly concentrated in the northern YRD region and the southeastern sector of Zhejiang Province. In contrast, 31.20% and 7.80% of the regions experience an increase or significant increase in average annual temperature, respectively, primarily situated in southern Anhui Province, southwestern Jiangsu Province, and the majority of northwestern Zhejiang Province. Temporally, the annual average temperature in the YRD region from 2000 to 2018 depicts a pattern of initial decrease followed by an upward trend, hitting the lowest annual average temperature of 16.08 °C in 2010 and reaching the highest annual average temperature of 16.84 °C in 2018 (Figure 8b).

Throughout the period from 2000 to 2018, the YRD region predominantly experiences a decrease or significant decrease in the average annual solar radiation trend, with rates of 67.90% and 18.67%, respectively. Conversely, only a minor 13.43% of the region exhibits an upward trajectory in average annual solar radiation, primarily found at the convergence of three provinces: southwestern Zhejiang Province, northern Anhui Province, and a small segment of northeastern Jiangsu Province. Temporally, the annual average solar radiation pattern across the overall region showcases a fluctuating sequence of increment, subsequent decline, and subsequent upsurge, culminating in the peak average solar radiation recorded in 2005 at 5243.70 MJ·m$^{-2}$ and the lowest level in 2015 at 4871.07 MJ·m$^{-2}$ (Figure 8c).

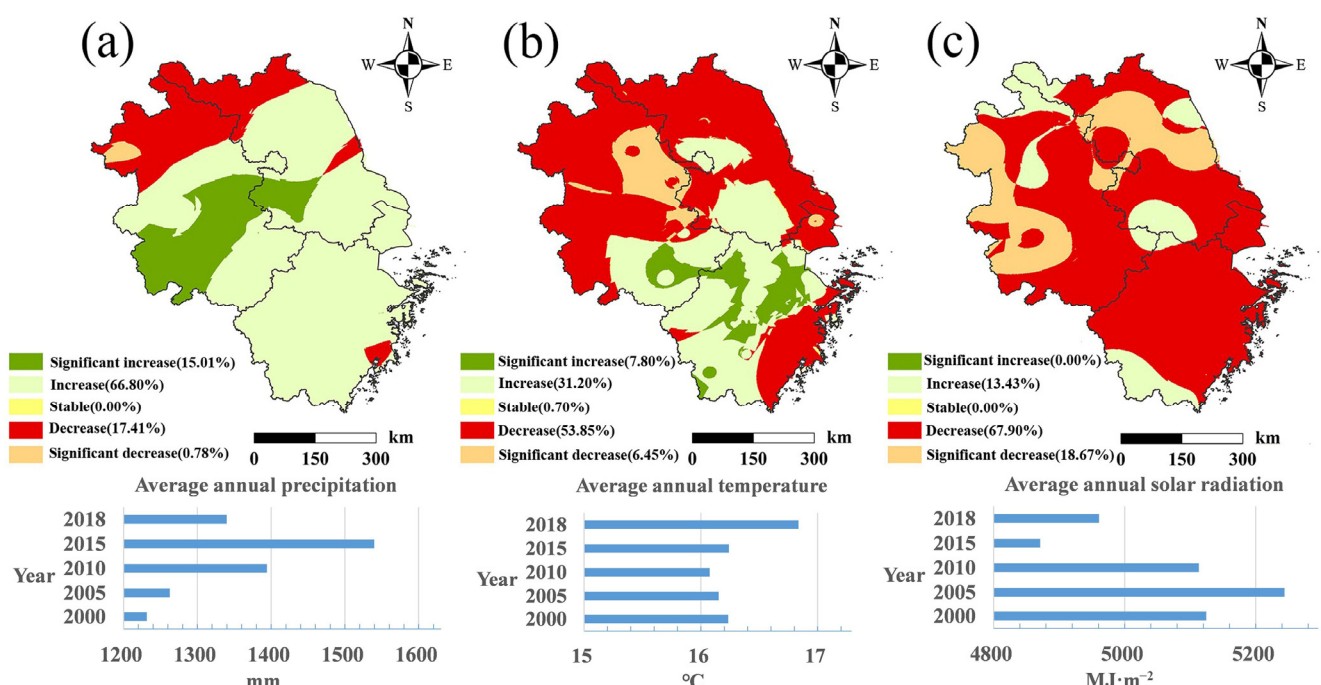

**Figure 8.** Changes in climatic factors in the YRD region from 2000 to 2018: (**a**) average annual precipitation, (**b**) average annual temperature, (**c**) average annual solar radiation.

3.4.2. Scenario Analysis: Climate Change Only

In the L2-4 scenarios, LULCC were maintained at the 2000 level, ensuring alterations solely in precipitation for the L2, temperature for the L3, and solar radiation for the L4 scenarios, subsequently examining the effects of climate change on NPP.

Initially, within the L2 scenario, it is evident that the average NPP in the YRD region follows a pattern of initial increase succeeded by a subsequent decline between 2000 and 2018. This pattern mirrors the trend observed in the realistic scenario post-2005 (Figure 9a). Solely influenced by precipitation changes, the average NPP rose from 2000 to 2015, peaking at 697.81 gC/(m²·a) in 2015, followed by a decrease to 689.25 gC/(m²·a) in 2018. Comparing these findings with the realistic scenario reveals a similar upward trajectory in NPP from 2005 to 2015, followed by a decline in 2018. Furthermore, in the L2 scenario, the average NPP consistently surpasses that of the realistic scenario across all years. This discrepancy suggests that the average NPP in the YRD region, influenced solely by precipitation changes, outperforms the NPP under the combined influence of multiple factors.

At the pixel scale, analyses reveal that NPP levels predominantly exhibit an increasing or notably rising trend in most regions under the singular influence of precipitation changes (L2). A minority of regions display a stable or declining trend, with none exhibiting a significant decrease (Figure 9a). Specifically, segments demonstrating a significant increase in NPP represent 24.40%, primarily clustered horizontally in the central segment of the YRD region, with additional distribution across various regions in the southern and central sectors of Zhejiang Province. Areas experiencing an upsurge in NPP comprise 73.07% and are extensively spread throughout the remaining YRD region. Conversely, regions displaying a decline in NPP make up 2.48%, primarily situated in the outer areas of the YRD region.

Secondly, within the L3 scenario, the average NPP in the YRD region exhibits a more pronounced pattern of fluctuation from 2000 to 2018 (Figure 9b). Under the influence of temperature changes alone, the average NPP value rose to 664.72 gC/(m²·a) in 2005 from 638.54 gC/(m²·a) in 2000. Subsequently declining to 639.98 gC/(m²·a) in 2010, it then surged to a peak of 685.58 gC/(m²·a) in 2015, before declining once more to 645.08 gC/(m²·a) in 2018. Moreover, the average NPP values across all years in the L3

scenario surpass those in the realistic scenario, signifying that NPP values are elevated solely under the impact of temperature when compared to the combined factors.

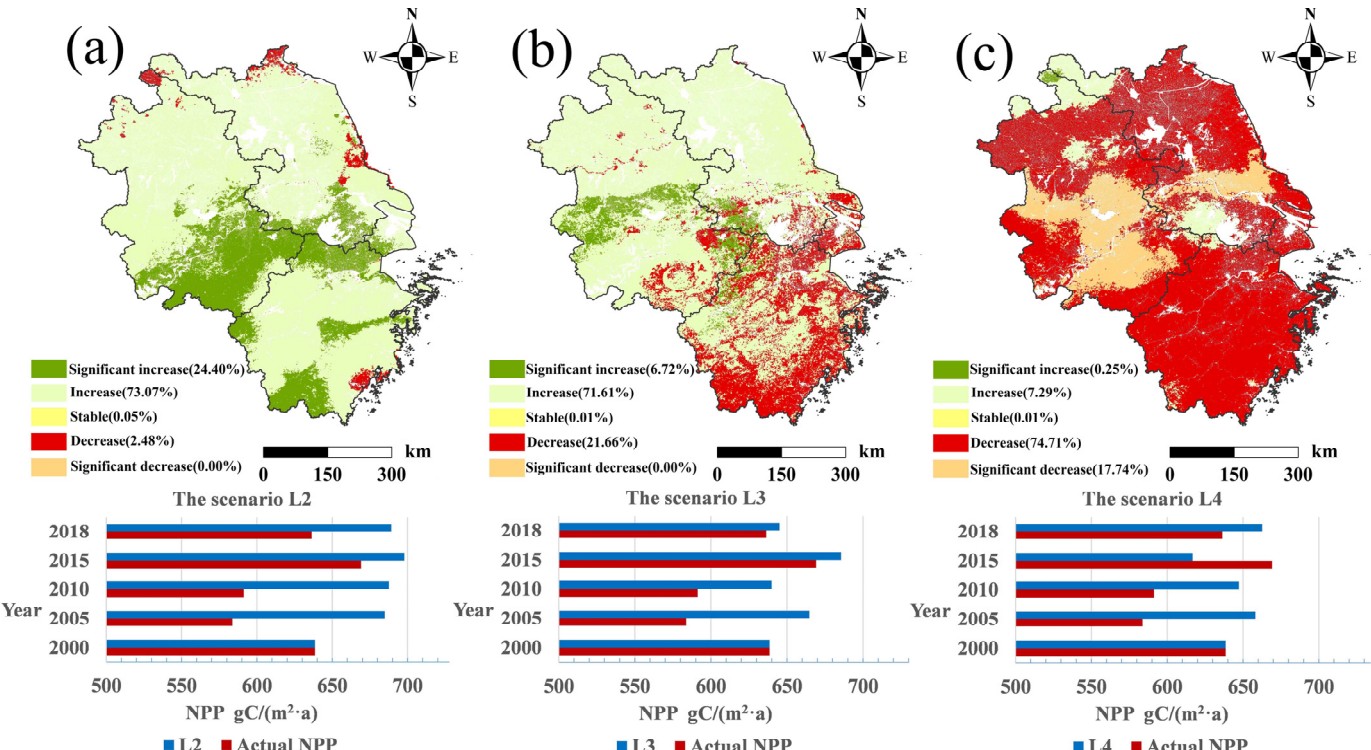

**Figure 9.** Changes in NPP in different scenarios: (**a**) L2, (**b**) L3, (**c**) L4.

At the pixel scale, trend analyses under the exclusive influence of temperature variations (L3) reveal that a majority of regions exhibit a rising trend in NPP (71.61%), with 6.72% demonstrating a notable increase. Nonetheless, 21.66% of regions display a declining trend in NPP changes. Geospatially, the NPP trend variations in the L3 scenario exhibit distinct north–south heterogeneity. Specifically, areas experiencing growth and notable enhancements in NPP are predominantly clustered in the northern YRD region, whereas regions displaying a declining trend in NPP are primarily situated in the central and southern segments of the YRD region (Figure 9b).

Thirdly, within the L3 scenario, the average NPP in the YRD region from 2000 to 2018 demonstrates notable fluctuation (Figure 9c). Solely influenced by solar radiation, the average NPP value rose to 658.18 gC/(m²·a) in 2005, declined to 616.75 gC/(m²·a) in 2015, then surged back to 662.59 gC/(m²·a) in 2018. A comparison with the realistic scenario indicates that the average NPP values surpass those in the realistic scenario in all years except 2015. This implies that, under the exclusive influence of solar radiation, the average NPP in 2015 is lower than the level under the combined factors, whereas the average NPP in other years exceeds the level under the combined factors.

At the pixel scale, the results of the trend analyses show that, due to changes in solar radiation (L4), the majority of regions exhibit a declining trend in NPP (74.71%), with 17.74% indicating a significant decrease. Fewer regions exhibit an increasing or significantly increasing trend in NPP, at 7.29% and 0.25%, respectively. Notably, a small portion of the northern YRD and the confluence of the three provinces displayed an increasing or significantly increasing trend in NPP. Whereas regions displaying a significant decrease in NPP are clustered laterally in the central YRD region, the majority of other regions exhibit a decline in NPP (Figure 9c).

## 4. Discussion

### 4.1. Spatiotemporal Dynamics of NPP

The results indicate that the NPP in the YRD region between 2000 and 2018 displayed distinct spatial differentiation characteristics, typically demonstrating a spatial distribution pattern with high values in the southern and western regions, and low values in the northern and eastern regions (Figure 3). This finding aligns with the research of Wang et al. [44]. It is hypothesized that the emergence of this spatial pattern is linked to the physical geographical features of the YRD region.

The southwestern region of the YRD is characterized by an abundance of mountain ranges, serving as a crucial natural resource foundation for the flourishing and progression of vegetation, along with fostering a resilient forest ecosystem. For instance, topographical elements can constrain alterations in land use and mitigate disruptions caused by human activities [45], while the growth parameters of vegetation show a progressive enhancement with rising altitudes within a specific threshold [46]. Moreover, owing to their intricate ecosystem arrangement and robust adaptability to climate variations, forests exhibit greater consistency in productivity and a broader spectrum of plant-derived carbon reserves compared to alternative ecosystems [47]. Consequently, clusters of high NPP values were prevalent across extensive regions here, resulting in the NPP levels in Anhui and Zhejiang provinces within the southwestern YRD region being relatively equivalent to or surpassing the average NPP level in the YRD region as a whole. Furthermore, a substantial portion of the areas that manifested notable rises in, or maintained an upward trajectory in, the mean NPP values were predominantly centered in this region. Additionally, it was observed that the region demonstrating elevated NPP values had seen a rise between 2005 and 2018, with Anhui Province and Zhejiang Province experiencing variations marked by fluctuating tendencies and positive escalations in the mean NPP values. This correlation could potentially be attributed to the ecological restoration initiatives undertaken by the Chinese government, including initiatives like land conversion from farmland to wooded areas and meadows, and forestation efforts on hill slopes, alongside strategies aimed at preserving water reservoirs [41]. These endeavors in ecological restoration serve to mitigate the influence of human interventions on vegetation, consequently enhancing the ecological milieu conducive for its prosperity [48].

The eastern and northern sectors of the YRD region offer conducive topographical features for human endeavors [49], and the NPP in these areas primarily remained at lower levels, notably in locales characterized by significant urbanization. Furthermore, our observations revealed a rising prevalence of regions exhibiting diminished NPP levels, often dispersed and expanding outwardly, mirroring the patterns associated with the sprawl of urbanized zones [50]. Consequently, the NPP in Shanghai and Jiangsu provinces, situated in the eastern and northern sectors of the YRD region, exhibited levels below the regional YRD average (Figure 4). Concurrently, a prevalent trend observed in the NPP dynamics across many of these areas involved a substantial decline, notably pronounced in southern Jiangsu, northern Zhejiang, and Shanghai, regions undergoing swift urbanization and high population densities (Figure 5). Furthermore, a noticeable proliferation of regions displaying diminished NPP levels was evident pre-2010, with Shanghai and Jiangsu witnessing a decline in mean NPP values within this timeframe, a trend that subsequently stabilized post-2010. This correlation could potentially be attributed to the strategic regional development initiatives of the Chinese government, exemplified by the notable emphasis on environmentally sustainable practices within the YRD, alongside escalated endeavors in ecological preservation post-2012 [51].

### 4.2. Impact of LULCC on NPP

Under the influence of LULCC alone (L1 scenario), the results indicate a substantial divergence in the change in NPP in the YRD region from 2000 to 2018 compared to the realistic scenario. After 2000, the NPP in the realistic scenario exhibited a fluctuating pattern of increase and decrease, whereas the NPP in the L1 scenario indicated a consistent

decrease (Figure 7a). This demonstrates the impact of LULCC on NPP [52] and confirms that changes in NPP are influenced by a combination of interconnected factors [24,25].

Regarding temporal changes, under the sole influence of LULCC, the YRD region exhibited a decline in the average NPP value (Figure 7a). Concurrently, the LULCC in the YRD region between 2000 and 2018 predominantly featured the expansion of built-up areas and the decrease in cultivated land (Table 6). As a result, the expansion of built-up areas and the reduction in cultivated land led to a decrease in NPP within the YRD region. This outcome aligns with the conclusions of Wu et al. [29] and Yang et al. [53], who observed that urbanization in the YRD region led to urban expansion and a decrease in arable land, adversely affecting NPP. This can be attributed to the rising population and rapid industrialization, which have increased the need for built-up spaces, leading to the conversion of extensive areas of soil covered with vegetation into impermeable surfaces. Such conversion directly or indirectly encroaches upon the crucial ecological space that vegetation depends on, thereby undermining its carbon sequestration capability [44].

In addition, it was observed that the conversion of cultivated land to built-up areas decelerated in 2015 (Table S3 in the Supplementary Materials) but remained the prevalent form of land-use transition (Figure 6), consequently resulting in a lower average NPP value in 2015 compared to the realistic scenario (Figure 7a). The net transformation of cultivated land to built-up areas was comparatively minimal in 2018, leading to a slightly lower average NPP for the same year compared to the realistic scenario (Figure 7a). All the aforementioned phenomena align with the adverse impact of built-up area expansion on NPP. However, the average NPP values in 2005 and 2010 were higher than the realistic scenario (Figure 7a), and the built-up area transferred within these two periods was still relatively large (Tables S1 and S2 in the Supplementary Materials). This may be due to the fact that the NPP in the L1 scenario reflects only the impact of LULCC, whereas the NPP in the realistic scenario is the result of the combined impact of multiple factors of LULCC and climate change, which in turn leads to higher levels of NPP than in the realistic scenario of these two periods. More specifically, firstly, climate change during these two periods may reduce the level of NPP in realistic scenarios. For example, the average annual precipitation was relatively lowest in 2005 (Figure 8a) and the average annual temperature was relatively lowest in 2010 (Figure 8b). Secondly, the impact of LULCC on NPP was relatively limited. In realistic scenarios, climate change may offset some of the positive impacts of LULCC on NPP. For example, while built-up areas were being transferred in 2005 and 2010, other LULCC, such as the conversion of cultivated land to forest and grassland to forest, may also had a positive impact on NPP. These resulted in higher NPP from the simulated scenarios and lower NPP from the realistic scenarios in 2005 and 2010.

In terms of space, under the impact of LULCC alone, only a small proportion of areas (9.41%) exhibit a declining trend in NPP (Figure 7b) and these areas are more similar to the areas where cultivated land was transferred to built-up areas in spatial distribution (Figure 6). This reinforces our assertion that the expansion of built-up areas will negatively affect NPP. We also found that 88.53% of the NPP exhibited an increasing trend of change (Figure 7b), and these areas have similar spatial distributions to other land type conversions and areas with stable land-use types, such as the conversion of cultivated land to forests (Figure 6). This further supports our view that other land-use transfers may also have a positive impact on NPP, specifically the conversion of other land types to forests. Furthermore, there are no regions of significant increase or decrease in NPP in the L1 scenario (Figure 7b), while significant increases or decreases in NPP occur in the L2-4 scenarios (Figure 9). We speculate that, although NPP is influenced by a combination of LULCC and climate factors, the impact of LULCC alone may be relatively limited, whereas climate change factors may have a more substantial impact on NPP, aligning with the findings of Yang et al. [52].

In conclusion, excluding the interference of climate change, our findings validate that LULCC affects NPP in the YRD region. Expansion of built-up areas and the reduction of cultivated land decreased NPP in the YRD region. Converting other land types to

forests could potentially benefit NPP. Furthermore, the impact of LULCC is expected to be relatively restricted in contrast to the effects of climate change.

### 4.3. Impact of Climate Change on NPP

Under the impact of climate change alone (L2-4 scenario), the results indicated that NPP in the YRD region from 2000 to 2018 exhibited distinct characteristics of change compared to the observed scenario. This finding supports the hypothesis that climate change can have an impact on NPP in the YRD region [29,54]. The YRD region is influenced both by sea–land thermal contrast and seasonal circulation; consequently, its ecosystems are highly sensitive to responses to climate change.

Firstly, considering only the impact of precipitation (L2), NPP levels in the YRD region were consistently higher than those in the realistic scenario in all years (Figure 9a). This implies that the mean NPP of the YRD region, when subject to solely precipitation changes, exceeds that under the combined influence of multiple factors. In conjunction with changes in precipitation, the average annual rainfall was relatively low in 2000 and 2005, then rose and reached a peak in 2015, before declining in 2018 (Figure 8a). Most areas within the L2 scenario experienced an increase or marked increase in NPP (Figure 9a), which was mirrored by a corresponding rise in average annual rainfall (Figure 8a). This is consistent with the findings of several studies that increased precipitation can promote NPP levels [29,54] and precipitation is the dominant variable affecting vegetation growth and its inter-annual variability [41,55]. Precipitation is the main source of water for the growth and development of vegetation, and the appropriate amount helps to maintain soil moisture, which is essential for the normal physiological activities of the crop root system [56]. Additionally, the nutrients contained in rainwater play an important role in the physiology and metabolism of vegetation, constituting an indispensable condition in the vegetative life system [57].

Secondly, when considering the influence of temperature only (L3), NPP levels in the YRD region consistently exceeded those of the realistic scenario for all years (Figure 9b). Notable spatial heterogeneity was detected in the association between NPP changes and the shifts in annual average temperature in the YRD region. In the L3 scenario, NPP in certain areas of the central and southern YRD region demonstrated an upward trend (Figure 9b), with the majority of these areas also experiencing an uptrend in average annual temperatures (Figure 8b). Simultaneously, the NPP in the southeastern coastal region exhibited a concurrent decline with temperature. This observation suggests that, in these regions, NPP changes were synergistically linked to temperature increases under the exclusive influence of temperature. Multiple studies have demonstrated an elevation in NPP associated with warming temperatures [5,54], and such a positive correlation was observed in segments of the central and southern YRD region. Concurrently, our study uncovered a deviating correlation in the northern YRD region, where NPP trends increased despite a decrease in temperature (Figures 7b and 8b). Wu et al. similarly found a negative correlation between NPP and temperature in parts of the YRD region [29], while Hao et al. also found such phenomena in parts of central China [55]. This may be due to the fact that higher temperatures increase the evapotranspiration rate of vegetation and soil, reducing soil moisture and enhancing vegetation respiration [47,58], which in turn limits seasonal vegetation growth [29]. At the same time, there may also be thresholds for changes in suitable temperatures [59], which can lead to a warming climate that begins to inhibit the carbon sequestration capacity of vegetation [60,61].

Finally, under the exclusive impact of changes in solar radiation (L4), the average NPP values in the YRD region also showed deviations from the realistic scenario. The variation characteristics of the annual average solar radiation were similar to those of vegetation NPP under the L4 scenario. We conjecture that there is a positive correlation between changes in solar radiation and vegetation NPP changes in the YRD region. In terms of the trend in solar radiation, most areas exhibited a decreasing or significantly decreasing trend (Figure 8c), and vegetation NPP in these areas also presented a decreasing or significantly decreasing

trend under the L4 scenario (Figure 9c). Spatial distribution similarities further confirm our conjecture. This aligns with the findings of several studies that suggest an increase in solar radiation can boost vegetation NPP levels [29,41]. Solar radiation is a necessary condition for vegetation to perform photosynthesis, and suitable sunlight promotes the effective absorption, transmission, and transformation of light energy by vegetation [62]. Provided other conditions remain constant, the photosynthetic capacity of vegetation may also increase with rising solar radiation [63].

In conclusion, upon excluding LULCC influences, our results demonstrate that climatic factors—precipitation, temperature, and solar radiation—influence NPP in the YRD region. Firstly, changes in precipitation displayed a positive correlation with NPP alterations, with the effect being markedly pronounced. Secondly, the correlation between temperature and NPP exhibited spatial variability, with a predominantly positive correlation in the central and southern regions of the YRD, and a predominantly negative correlation in the northern region. Thirdly, changes in solar radiation were negatively correlated with NPP fluctuations.

### 4.4. Policy Implications

To enhance NPP levels in the YRD region, we offer the following targeted recommendations, considering the dynamic nature of LULCC and climatic factors. (1) Enforce urban development boundaries rigorously and restrain the disorderly sprawl of built-up areas. In the context of LULCC in the YRD region, Jiangsu Province witnessed the largest net conversion of cultivated land, particularly in the southern part where urban area expansion was more marked. Furthermore, similar urban encroachments upon agricultural lands occur in the eastern coastal regions of Zhejiang Province and Shanghai (Figure 6). Therefore, it is imperative to adhere to the designated boundaries of permanent farmland and urban zones strictly during urban construction to avoid the ungoverned sprawl of urban areas encroaching on arable land. Concurrently, enhancing the management of inefficient urban spaces and intensifying re-greening initiatives is crucial to reduce the detrimental effects of human activities on NPP. (2) Upgrade the regional irrigation systems and tackle air pollution to secure the essential conditions for vegetation growth. Observations indicate a diminishing trend in precipitation in northern Jiangsu Province, and a similar pattern, from moderate to significant reductions, in northern Anhui Province (Figure 8a). As such, intervention via artificial precipitation in areas of high vegetation density like croplands and forests is warranted in these regions. Additionally, real-time monitoring of soil moisture and vegetation growth is imperative, as is the renewal and upkeep of regional irrigation infrastructures. Establishing a scientific water resource allocation system, strengthening rainwater harvesting techniques, and tailoring water supply to specific needs based on water demand, soil moisture, and vegetation type are all strategies that could provide a robust foundation for the growth and development of vegetation. Most of the YRD region exhibited a decreasing trend in solar radiation (Figure 8c); thus, controlling atmospheric pollution, such as industrial emissions, is essential to reduce the reflection and scattering of solar radiation by aerosols, ensuring adequate sunlight for vegetation, which is vital for growth and photosynthesis. (3) Reduce greenhouse gas emissions and control regional temperature increases. The areas experiencing increased or significantly increased temperatures were primarily located in southern Anhui Province, southern Jiangsu Province, and northern Zhejiang Province (Figure 8b), which are more developed in terms of urbanization, industrialization, and population density. The urban heat island effect is a likely contributor to higher temperatures in these regions. Adjusting and optimizing the industrial energy structure to lessen reliance on fossil fuels and enhancing the efficiency and affordability of clean energy usage are strategic priorities.

### 4.5. Limitations and Future Perspectives

Scenario analysis has enabled the exploration of the impacts of LULCC and climate change on NPP in the YRD region. This approach facilitated the control of impact conditions

and the exclusion of confounding factors, providing a stronger scientific basis for elucidating the impact mechanisms of NPP. However, this study did not examine the collective impact of LULCC and climatic factors on NPP. Future research should expand the scenario analysis to explore the interplay among the influencing factors affecting NPP. Additionally, this study validated the CASA model's measurements through comparative analysis but did not provide ground validation results. Validation methods could be diversified in future research.

**5. Conclusions**

In this study, the NPP of the YRD region between 2000 and 2018 was quantified using the CASA model. The spatiotemporal dynamics and influencing factors of NPP were investigated using Theil–Sen median trend analysis and scenario analysis. The results revealed the following key findings: (1) The NPP in the YRD region between 2000 and 2018 exhibited distinct spatial differentiation characteristics, generally following a pattern of higher values in the southern and western areas, and lower values in the northern and eastern parts. Areas with lower NPP levels showed an increasing trend, often expanding outward, similar to the growth pattern of built-up areas. For example, NPP levels in Shanghai and Jiangsu provinces, which are located in the eastern and northern regions of the YRD, were below the regional average. Conversely, areas with higher NPP levels were concentrated in large parts of the southwestern region of the YRD, with an overall increasing trend. As a result, Anhui and Zhejiang provinces in the southwest YRD region generally had NPP levels equal to or higher than the regional average. (2) The expansion of built-up areas and the reduction in cultivated land were associated with lower NPP levels. Meanwhile, the conversion of non-forest land types to forest showed a positive effect on NPP. Nevertheless, the effect of LULCC was found to be relatively limited compared to that of climate change. (3) Our study found that climatic factors such as precipitation, temperature, and solar radiation significantly affected NPP in the YRD region. Changes in precipitation showed a positive correlation with changes in NPP, with a more pronounced effect. Temperature–NPP correlations varied spatially, showing predominantly positive relationships in the central and southern regions of the YRD, but predominantly negative relationships in the northern zone. In addition, changes in solar radiation were negatively correlated with changes in NPP.

**Supplementary Materials:** The following supporting information can be downloaded at: https://www.mdpi.com/article/10.3390/land13040440/s1, Table S1: Land-use transfer matrix in YRD region during 2000–2005 ($km^2$); Table S2: Land-use transfer matrix in YRD region during 2005–2010 ($km^2$); Table S3: Land-use transfer matrix in YRD region during 2010–2015 ($km^2$); Table S4: Land-use transfer matrix in YRD region during 2015–2018 ($km^2$).

**Author Contributions:** Conceptualization, Q.F. and M.G.; methodology, Q.F. and M.G.; software, M.G.; formal analysis, T.W.; resources, Q.F. and J.C.; data curation, T.W.; writing—original draft preparation, Q.F. and M.G.; writing—review and editing, Q.F. and T.W.; visualization, Q.F.; supervision, J.C.; project administration, Q.F.; funding acquisition, Q.F. All authors have read and agreed to the published version of the manuscript.

**Funding:** This research was funded by the National Natural Science Foundation of China (42101253) and the Jiangsu Province Social Sciences Application Research Boutique Project (23SYC-180).

**Data Availability Statement:** The original contributions presented in the study are included in the article and Supplementary Materials, further inquiries can be directed to the corresponding authors.

**Acknowledgments:** We would like to thank the anonymous reviewers for their valuable comments and suggestions.

**Conflicts of Interest:** The authors declare no conflicts of interest.

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
