# Peer review of "Spatiotemporal Dynamics and Influencing Factors of Vegetation Net Primary Productivity in the Yangtze River Delta Region, China"

_land, doi:10.3390/land13040440_

Round 1

Reviewer 1 Report

Comments and Suggestions for Authors

1. In the abstract, I propose to set out the significance of the research.

2. In the abstract, after the words "The results show that", there is only "(1)", and I suggest that "(1)" be deleted.

3. At the end of the introduction, after "(3)", I suggest a separate paragraph.

4. I suggest deepening the colour of figure 3

5. I'd suggest making Figure 4 smaller.

6. The theoretical contribution and practical significance of the paper are not very clear

7. Whether there are any limitations to the research in this paper and, if so, what they are.

8. I suggest that an appropriate outlook could be added

Comments on the Quality of English Language

The language of the article should be more standardised.

Author Response

We are extremely grateful to you for reviewing our manuscript and providing comprehensive comments. Your comments and suggestions are invaluable in guiding the revision and enhancement of our paper, and hold significant importance for our research. We have carefully reviewed the comments and made corrections, which we hope will meet with your approval. The revised sections were highlighted in red within the document. Detailed below are the principal corrections made to the paper, alongside responses to the reviewer's comments:

Reviewer 1:

Comments and Suggestions for Authors

Comment 1: In the abstract, I propose to set out the significance of the research.

Response to comment 1: We would like to express our sincere thanks to you for your comments. We have emphasized the significance of the research in the abstract (page 1, lines 33-36). The specific modifications are detailed as follows:

This study can provide stronger evidence for revealing the influencing mechanisms of NPP through the control of impact conditions and the exclusion of confounding factors via scenario analysis. The policy implications can offer insights into NPP enhancement and environmental management for the YRD and other rapidly urbanizing regions.

Comment 2: In the abstract, after the words "The results show that", there is only "(1)", and I suggest that "(1)" be deleted.

Response to comment 2: Thank you for pointing out the error. We sincerely apologize for the error in the manuscript. We have addressed the pertinent issues (page 1, line 19) and conducted a thorough proofreading of the manuscript.

Comment 3: At the end of the introduction, after "(3)", I suggest a separate paragraph.

Comment 6: The theoretical contribution and practical significance of the paper are not very clear.

Response to comment 3 and 6: Thank you for your comments. We are sorry that we did not express clearly the theoretical contribution and practical significance of this study. After the main research objectives of this paper, we have added a paragraph and highlighted the theoretical contribution and practical significance (page 3, lines 112-117). The specific modifications are as follows:

This study analyses the impact of LULCC and climatic factors on NPP by the control of impact conditions and the exclusion of confounding factors through scenario analysis, so as to provide stronger evidence for revealing the influencing mechanism of NPP. The practical value is to provide the spatiotemporal dynamics of NPP in the YRD region and the policy implications for enhancing the level of NPP, which in turn serves as a reference for environmental management in the YRD region and other rapidly urbanizing regions.

Comment 4: I suggest deepening the colour of figure 3.

Response to comment 4: Thank you for your suggestion. We have deepened the colour of this figure (page 11, line 345).

Comment 5: I'd suggest making Figure 4 smaller.

Response to comment 5: Thank you for your suggestion. We have made this figure smaller (page 11, line 357)

Comment 7: Whether there are any limitations to the research in this paper and, if so, what they are.

Comment 8: I suggest that an appropriate outlook could be added.

Response to comment 7 and 8: Thank you for your comments. We have illustrated the limitations of the research and provided the outlook for future research (page 21, lines 745-753). The details are as follows:

Scenario analysis has enabled the exploration of the impacts of LULCC and climate change on NPP in the YRD region. This approach facilitated the control of impact conditions and the exclusion of confounding factors, providing a stronger scientific basis for elucidating the impact mechanisms of NPP. However, this study did not examine the collective impact of LULCC and climatic factors on NPP. Future research should expand the scenario analysis to explore the interplay among influencing factors affecting NPP. Additionally, this study validated the CASA model's measurements through comparative analysis but did not provide ground validation results. Validation methods could be diversified in future research.

Comments on the Quality of English Language: The language of the article should be more standardised.

Response to this comment: Thank you for your suggestion. Native English speakers have helped us polish the paper.

We tried our best to improve the manuscript and made some changes in the manuscript. These changes will not influence the content and framework of the paper. We appreciate for your warm work earnestly and hope that the correction will meet with approval. Once again, thank you very much for your comments and suggestions.

Best regards,

Tinghui Wang, Qi Fu, and Jinhua Chen

Reviewer 2 Report

Comments and Suggestions for Authors

I am happy to review the manuscript. Authors illustrated the spatiotemporal variation of NPP using CASA method with their influencing factors such as LULCC, meteorological and climatic parameters, and validated with other methods for assessing NPP. I have some overall comments as below:

·       Abstract section is quite clear to understand what they wanted to do in their research and well organized.

·       Authors introduce well about the importance of the work with relevant literature and objectives of the study.

·       The abbreviation of Land use and land cover changes should be like this ‘LULCC’ and it must address first before using the abbreviation. There is no need to mention the abbreviation repetitively.

·       A flow chart of the research could be address to clearly understand the methodology.

·       Ground validation is missing in this study, although the authors validated their model outcome with other related model’s study.

·       Result section is quite good and illustrated well.

·       They discussed their results with relevant information and suggested some recommendation for enhancing NPP for the policy makers.

Overall, the manuscript looks good and suggested to accept the manuscript with addressing the above mentioning comments.

Author Response

We are extremely grateful to you for reviewing our manuscript and providing comprehensive comments. Your comments and suggestions are invaluable in guiding the revision and enhancement of our paper, and hold significant importance for our research. We have carefully reviewed the comments and made corrections, which we hope will meet with your approval. The revised sections were highlighted in red within the document. Detailed below are the principal corrections made to the paper, alongside responses to the reviewer's comments:

Reviewer 2:

I am happy to review the manuscript. Authors illustrated the spatiotemporal variation of NPP using CASA method with their influencing factors such as LULCC, meteorological and climatic parameters, and validated with other methods for assessing NPP. I have some overall comments as below:

Comment 1: Abstract section is quite clear to understand what they wanted to do in their research and well organized.

Comment 2: Authors introduce well about the importance of the work with relevant literature and objectives of the study.

Response to comment 1 and 2: We appreciate your valuable feedback on our abstract and introduction sections. Your encouraging comments motivate us to continually strive for clarity and coherence in our manuscript.

Comment 3: The abbreviation of Land use and land cover changes should be like this ‘LULCC’ and it must address first before using the abbreviation. There is no need to mention the abbreviation repetitively.

Response to comment 3: Thank you for your correction. We are very apologetic about the error that appeared in the manuscript. We have corrected the relevant problems (page 1, lines 23-24 and page 2, lines 78-79) and re-proofread the manuscript.

Comment 4: A flow chart of the research could be address to clearly understand the methodology.

Response to comment 4: Thank you for this comment. We have added a paragraph on the analytical workflow for this study (page 5, lines 169-179) and provided a flow chart of the research (page 5, Figure 2). The specific modifications are as follows:

The analytical workflow of this study comprised three main steps. Initially, NPP was quantified using the CASA model, with results validated against related studies and MOD17A3 data. Subsequently, the spatial distribution and dynamic trends of NPP were investigated through Trend Analysis. Secondly, the spatiotemporal variations in LULCC and climate factors, including precipitation, temperature, and solar radiation, were analyzed. Concurrently, four simulated scenarios were established to investigate the effects of various factors on NPP using Trend Analysis. Thirdly, the NPP changes in the simulated scenarios were compared to the realistic scenario. By combining the spatiotemporal variations of LULCC and climate factors, the impacts of LULCC and climate change on NPP were discussed. Based on this analysis, policy implications for enhancing NPP levels in the YRD region were suggested. The flow chart of this research is displayed in Figure 2.

Comment 5: Ground validation is missing in this study, although the authors validated their model outcome with other related model’s study.

Response to comment 5: Thank you for your comment. We appreciate and fully agree that utilizing field data collection is a valuable method for validating the CASA model. Following your suggestion, we have outlined some research plans. Field data collection forms the foundation for ground validation, with sample methods and spectrometers being two conventional and widely used approaches. Our plan involves gathering ground data, including vegetation species count, species diversity, soil composition, and vegetation cover, through these methods. Once sample zones and plots in the YRD are identified, the plots will be photographed, and measurements of plot coverage and heights of various species will be recorded. Concurrently, plant samples and their fresh and dry weights will be obtained. The spectrometer will be calibrated, checking parameters like spectral resolution, central wavelength, and signal-to-noise ratio. Data collection will be conducted in a representative area under suitable weather conditions and timeframes. Subsequently, mathematical and statistical methods will be applied to analyze the spectral characteristics and derive remotely sensed vegetation parameters, recording information such as latitude, longitude coordinates, and spectral identifiers of the samples. This data will be used to extract characteristic parameters like vegetation reflectance and identify vegetation cover, enabling the estimation of NPP in the YRD region.

However, as our study focuses on the dynamic NPP processes in the YRD from 2000 to 2018 and the underlying influencing factors, the current field data collection is not sufficient for validating long time series. Hence, we opted for comparative validation with existing studies and remote sensing data in this paper to access validation data over an extended period. While the ground validation method demands substantial equipment preparation and processing and does not align precisely in the temporal dimension, we regret that we are unable to employ this method in the present paper. Nevertheless, your suggestion has pointed us towards a valuable direction for future research. We have acknowledged and addressed this limitation in our paper and commit to enhancing the CASA model validation method in our subsequent research (page 21, lines 750-753).

Comment 6: Result section is quite good and illustrated well.

Comment 7: They discussed their results with relevant information and suggested some recommendation for enhancing NPP for the policy makers.

Response to comment 6 and 7: Thank you for your insightful comments on our manuscript. We are pleased to receive your positive feedback on the result and discussion section. It is our goal to not only present our findings but also to offer practical suggestions that can contribute to environmental policy decisions. Your feedback is invaluable to us, and it motivates us to continue striving for excellence in our work.

We tried our best to improve the manuscript and made some changes in the manuscript. These changes will not influence the content and framework of the paper. We appreciate for your warm work earnestly and hope that the correction will meet with approval. Once again, thank you very much for your comments and suggestions.

Best regards,

Tinghui Wang, Qi Fu, and Jinhua Chen

Reviewer 3 Report

Comments and Suggestions for Authors

Title: Spatiotemporal Evolution and Influencing Factors of Vegetation Net Primary Productivity in the Yangtze River Delta Region, China.

Research is conducted to evaluate the net primary productivity (NPP) employing CASA model in the delta region of China. Research design is strong, conducted and presented well and can be considered with minor edits

1. In the methodology section, the procedural steps of NPP calculation, the calculation of FPAR seems to missing.

2. Results are well presented and explained.

3. Discussion is well explained but a little lengthy. can be considered more concise.

Overall, research is conducted well. 

Author Response

We are extremely grateful to you for reviewing our manuscript and providing comprehensive comments. Your comments and suggestions are invaluable in guiding the revision and enhancement of our paper, and hold significant importance for our research. We have carefully reviewed the comments and made corrections, which we hope will meet with your approval. The revised sections were highlighted in red within the document. Detailed below are the principal corrections made to the paper, alongside responses to the reviewer's comments:

Reviewer 3:

Research is conducted to evaluate the net primary productivity (NPP) employing CASA model in the delta region of China. Research design is strong, conducted and presented well and can be considered with minor edits. Overall, research is conducted well.

Comment 1: In the methodology section, the procedural steps of NPP calculation, the calculation of FPAR seems to missing.

Response to comment 1: Thank you for your comment. We apologize for not providing the FPAR calculation steps in the article. We have added the formulas and presentations in the article (page 6, lines 210-225). 

Comment 2: Results are well presented and explained.

Response to comment 2: Thank you for your insightful comments on our manuscript. We are pleased to receive your positive feedback on the result section. Your feedback is invaluable to us, and it motivates us to continue striving for excellence in our work.

Comment 3: Discussion is well explained but a little lengthy. can be considered more concise.

Response to comment 3: We appreciate your valuable feedback on our discussion section. We have carefully reviewed the discussion section with the aim of streamlining it appropriately. However, the elaboration of the discussion section is closely linked to the results section. Our results provide a lot of relevant information and so it needs to be analyzed and explored in the discussion section. Therefore, unfortunately, it is very difficult to weed out some of the discussion. In any case, we thank you very much, as your suggestion provides a very valuable guideline for our future research.

We tried our best to improve the manuscript and made some changes in the manuscript. These changes will not influence the content and framework of the paper. We appreciate for your warm work earnestly and hope that the correction will meet with approval. Once again, thank you very much for your comments and suggestions.

Best regards,

Tinghui Wang, Qi Fu, and Jinhua Chen

Reviewer 4 Report

Comments and Suggestions for Authors

Manuscript ID_ Land-2927822 with Title: Spatiotemporal Evolution and Influencing Factors of Vegetation Net Primary Productivity in the Yangtze River Delta Region, China.

The manuscript is interesting and well-prepared. It analyzes and discusses a critical issue connected with the quality of living space for millions of people. I only have some minor comments.

I suggest the authors avoid the word evolution and replace it with dynamic (or maybe development) throughout the text, in the title and keywords. The topic they analyze is the spatiotemporal dynamic or spatiotemporal development of vegetation NPP.

So, I recommend the title: Spatiotemporal Dynamic and Influencing Factors of Vegetation Net Primary Productivity in the Yangtze River Delta Region, China.

Please also explain the abbreviation LUCC in the abstract.

Figure 1: I find five maps of land use types redundant. Please keep only two, b) and f), displaying the situation in 2000 and 2018.

Figure 2: I am unsure if the readers will be able to read and compare all five sub-figures. I recommend keeping a) and e) and enlarging these two figures with lines displaying borders between provinces.

Figures 7 and 8: I suggest enlarging these two compound Figures to the whole page width. 

Author Response

We are extremely grateful to you for reviewing our manuscript and providing comprehensive comments. Your comments and suggestions are invaluable in guiding the revision and enhancement of our paper, and hold significant importance for our research. We have carefully reviewed the comments and made corrections, which we hope will meet with your approval. The revised sections were highlighted in red within the document. Detailed below are the principal corrections made to the paper, alongside responses to the reviewer's comments:

Reviewer 4:

The manuscript is interesting and well-prepared. It analyzes and discusses a critical issue connected with the quality of living space for millions of people. I only have some minor comments.

Comment 1: I suggest the authors avoid the word evolution and replace it with dynamic (or maybe development) throughout the text, in the title and keywords. The topic they analyze is the spatiotemporal dynamic or spatiotemporal development of vegetation NPP.

Comment 2: So, I recommend the title: Spatiotemporal Dynamic and Influencing Factors of Vegetation Net Primary Productivity in the Yangtze River Delta Region, China.

Response to comment 1 and 2: Thank you very much for your suggestions. We totally agree with you and have replaced "evolution" with " dynamic" throughout the manuscript and changed the title.

Comment 3: Please also explain the abbreviation LUCC in the abstract.

Response to comment 3: Thank you for your comment. We have explained the relevant abbreviation (page 1, lines 23-24) and corrected it to LULCC.

Comment 4: Figure 1: I find five maps of land use types redundant. Please keep only two, b) and f), displaying the situation in 2000 and 2018.

Response to comment 4: Thank you for this comment. We have removed the redundant images and kept only the years 2000 and 2018 (page 4, line 146). 

Comment 5: Figure 2: I am unsure if the readers will be able to read and compare all five sub-figures. I recommend keeping a) and e) and enlarging these two figures with lines displaying borders between provinces.

Response to comment 5: Thank you for your suggestion. We have enlarged this figure and deepened the boundaries between provinces (page 10, line 328). However, there is an analysis of temporal changes in NPP for 2005, 2010, and 2015 (page 9, lines 319-327), as well as a related discussion based on this result. Therefore, we have retained the spatial distribution of NPP for the intermediate years. The modified figure is as follows:

Figure 3. Spatial distribution of NPP in the YRD region from 2000 to 2018: (a) 2000, (b) 2005, (c) 2010, (d) 2015, (e) 2018.

Comment 6: Figures 7 and 8: I suggest enlarging these two compound Figures to the whole page width.

Response to comment 6: Thank you for your comment. We have enlarged these two compound figures (page 15, line 474 and page 17, line 533).

We tried our best to improve the manuscript and made some changes in the manuscript. These changes will not influence the content and framework of the paper. We appreciate for your warm work earnestly and hope that the correction will meet with approval. Once again, thank you very much for your comments and suggestions.

Best regards,

Tinghui Wang, Qi Fu, and Jinhua Chen
